# SONAR: Spectral-Contrastive Audio Residuals for Robust Deepfake Detection

## Abstract

Deepfake (DF) audio detectors still struggle to generalize to out of distribution inputs. A central reason is *spectral bias*, the tendency of neural networks to learn low-frequency structure before high-frequency (HF) details, which both causes DF generators to leave HF artifacts and leaves those same artifacts under-exploited by common detectors. To address this gap, we propose **Spectral-cONtrastive Audio Residuals (SONAR)**, a frequency-guided framework that explicitly disentangles an audio signal into complementary representations. An XLSR encoder captures the dominant low-frequency content, while the same cloned path, preceded by learnable SRM, value-constrained high-pass filters, distills faint HF residuals. Frequency cross-attention reunites the two views for long- and short-range frequency dependencies, and a frequency-aware Jensen–Shannon contrastive loss pulls real content–noise pairs together while pushing fake embeddings apart, accelerating optimization and sharpening decision boundaries. Evaluated on the ASVspoof 2021 and in-the-wild benchmarks, SONAR attains state-of-the-art performance and converges four times faster than strong baselines. By elevating faint high-frequency residuals to first-class learning signals, SONAR unveils a fully data-driven, frequency-guided contrastive framework that splits the latent space into two disjoint manifolds: natural-HF for genuine audio and distorted-HF for synthetic audio, thereby sharpening decision boundaries. Because the scheme operates purely at the representation level, it is architecture-agnostic and, in future work, can be seamlessly integrated into any model or modality where subtle high-frequency cues are decisive.

## 1 Introduction

**Why deepfake detection matters.** Generative AI now enables the creation of photorealistic images, video, and speech. In 2024, political deepfakes flooded social media during global elections, while voice-cloning scams caused multimillion-dollar losses, including a 25M$ transfer United Nations Development Programme (2024); TRM Labs (2025). The FBI warns of AI-powered vishing Cybersecurity Dive (2025). More broadly, synthetic media erodes trust in journalism, markets, and legal evidence, making robust detection essential.

**Prior work, and why it falls short.** Most forensic research still centers on ever-deeper classifiers, overlooking how deepfake artifacts disturb the *joint* statistics of content and noise. Early SRM-style detectors either use fixed high pass filters Fridrich & Kodovský (2012); Qian et al. (2020) or, in the case of Bayar & Stamm's constrained convolution Bayar & Stamm (2016), *learnable* prediction-error kernels that deliberately suppress content. Yet all of these methods operate on high frequency (HF) residuals *in isolation*, ignoring their correlation with the underlying signal. Han et al. Han et al. (2021) add a second, content branch with learnable SRM filters, but the two streams are only fused at the top and no constraint enforces statistical coupling, Zhu et al. Zhu et al. (2024) similarly boost noise for image forgery detection, treating it as an auxiliary cue that still requires pixel-level masks. None of these approaches capture the higher-order dependency between HF noise and semantic content an interplay that isolated filtering or late fusion cannot model.

In audio forensics, the field has progressed from handcrafted spectral features combined with GMM or LCNN classifiers Yamagishi et al. (2019) to fine-tuned self-supervised encoders such as HuBERT

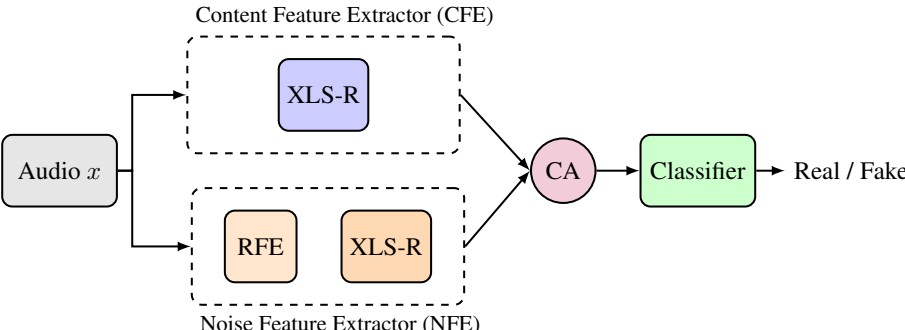

Figure 1: **SONAR overview.** Audio is processed in parallel by the Content Feature Extractor (CFE) and the Noise Feature Extractor (NFE). Their embeddings are fused via cross-attention (CA) and classified as real/fake.

and XLSR Hsu et al. (2021); Tak et al. (2022b); Zhang et al. (2024); Xiao & Das (2024). However, frequency-aware approaches have not yet been explored in this domain. Moreover, neither line of work models the alignment between semantic content and high frequency noise.

We state that all the previous work both in images and in audio suffers from a common limitation: **spectral bias**, also known as the *frequency principle* (F-principle) Rahaman et al. (2019); Basri et al. (2019); Cao et al. (2019); Xu et al. (2024); Fridovich-Keil et al. (2022). Whereby deep networks favor low frequency structure first and leave subtle HF cues under-represented. Although image methods partly address this by routing HF residuals through a separate branch, they stop at *mere separation*, they never model how low and high frequency information *should co-vary* in the same feature space during the learning process. There is no existing detector, visual or auditory that actively *aligns* genuine content–noise pairs while *repelling* their fake counterparts in latent space. This *alignment gap* is especially detrimental in audio, where high frequency artifacts are easily masked by perceptual post-processing, and, to date, no audio (or image) deepfake detector has explicitly addressed this issue in a data-driven application.

To better understand these limitations and motivate our solution, we conducted an exploratory statistical analysis of real vs. fake utterances across our train and test datasets. Our findings confirm that deepfakes differ from real audio not only in energy, but also in higher-order frequency statistics. Specifically, we observed a breakdown of natural low↔high co-modulation patterns (Fig. 2a), together with systematic shift in the contrast between low- and high-frequency energy bands (Fig. 2b). These cues cannot be exploited by fixed filters alone and strongly justify SONAR's use of learnable, distribution-level alignment.

**Our approach: SONAR and the gap it fills.** We close this gap with **SONAR**, a frequency guided, dual path framework that **learns** a bank of *data-driven SRM filters* to isolate high frequency (HF) residuals and imposes a Jensen–Shannon divergence loss to *pull* real content–noise pairs together while *pushing* fake pairs apart in latent space. By transforming HF residuals from a nuisance into a supervisory signal and learning their alignment with semantic content in feature space, SONAR directly combats spectral bias, accelerates convergence, and sets new state-of-the-art performance on both controlled benchmarks (ASVspoof 2021) and challenging "in the wild" audio. To our knowledge, it is the *first* audio deepfake detector to exploit **learnable, distributional alignment** between low and high frequency embeddings.

**Our contributions are threefold.**

- **SONAR: frequency-contrastive dual path.** We introduce **SONAR**, the first audio deepfake detector to jointly model low-frequency content and high-frequency residuals, turning spectral bias into a discriminative signal.

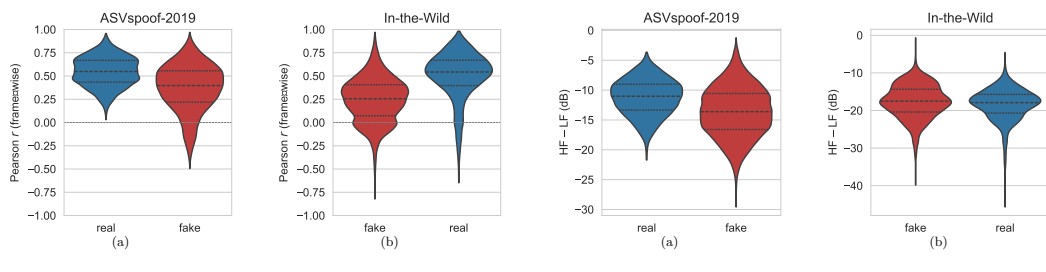

(a) LF–HF correlation.                    (b) HF/LF energy contrast.

Figure 2: **Low–high frequency structure reveals spoofing artifacts.** (a) Pearson correlation between low- (0–4 kHz) and high-frequency (7–8 kHz) bands shows real speech with strong co-modulation ($r \approx 0.6$), while fakes collapse toward zero or negative values. (b) The energy difference $\Delta E = E_{\mathrm{HF}} - E_{\mathrm{LF}}$ is systematically shifted for fakes across corpora, exposing a consistent HF/LF imbalance. These second-order cues motivate SONAR's *distributional alignment* objective.

- **Learnable SRM & JS frequency alignment loss.** A learnable SRM filter bank with a Jensen–Shannon alignment loss explicitly aligns real content–noise pairs and separates fake ones, a formulation not used in prior work.

- **State-of-the-art with fast convergence.** SONAR achieves new SOTA EERs on ASVspoof 2021 and In-the-Wild, converging in as few as 12 epochs while remaining robust to codecs and bandwidth shifts.

## 2 RELATED WORK

**High frequency cues in deep learning.** Fourier features markedly reduce spectral bias in MLPs Tancik et al. (2020). Successors such as Wave NN Yang et al. (2022), BiHPF Jeong et al. (2022), and ADD Woo (2022) insert explicit high pass branches or filters, showing that frequency-aware modules consistently sharpen detail capture.

**Frequency domain image forgery detection.** Two stream, high pass pipelines detect manipulation artifacts by pairing low pass content with residual branches Masi et al. (2020); Qian et al. (2020); Bayar & Stamm (2016); Fridrich & Kodovský (2012). Denoising-guided schemes Zhu et al. (2024) and compact frequency blocks Tan et al. (2024) further improve generalization with fewer parameters.

**Audio deepfake detection.** Classic systems combine handcrafted cepstral features with GMM/LCNN backends Yamagishi et al. (2019). Modern approaches leverage SSL encoders (Hu-BERT, Wav2Vec, XLSR, Whisper, WavLM) Hsu et al. (2021); Baevski et al. (2020); Babu et al. (2021); Radford et al. (2022); Chen et al. (2022), yet often falter on out of distribution (OOD) audio Müller et al. (2022). Tak *et al.* fine tuned XLSR with an AASIST head and augmentation for strong OOD results Tak et al. (2022b), later work fused XLSR layers with specialized classifiers to push performance further Zhang et al. (2024); Truong et al. (2024); Xiao & Das (2024).

**Our contribution.** Building on Tak et al. (2022b) and Xiao & Das (2024), we add a *learnable* dual path filter that explicitly aligns content and noise embeddings, boosting sensitivity to subtle high frequency artifacts. The result is faster convergence and state-of-the-art robustness across both benchmark and in the wild tests.

## 3 MATHEMATICAL MOTIVATION

**Spectral bias as a *coupling* defect.** A spoken frame $X$ contains *low-frequency* formants $L$ and *high frequency* micro-structure $H$, produced by the *same* vocal–tract event, hence jointly distributed: $p_{\mathrm{real}}(L, H) \neq p(L)\, p(H)$. Deep learning training is **spectrally biased**: it first fits the high energy LF error then runs out of budget, leaving an HF "hole" Rahaman et al. (2019). This provides us motivation for the assumption of approximate factorization $p_{\mathrm{fake}}(L, H) \simeq p(L)\, p(H)$ and, critically,

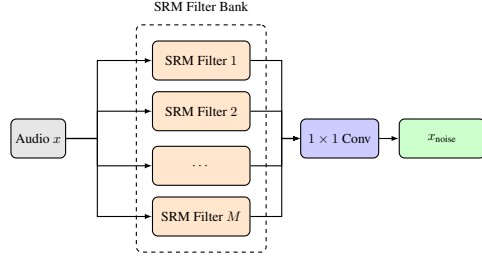

Figure 3: **Rich Feature Extractor (RFE).** Audio $x$ is processed by a bank of $M$ SRM-inspired filters, concatenated, and passed through a $1 \times 1$ learnable convolution layer to produce the noise residual representation $x_{\text{noise}}$.

a *mismatch* in the *joint* LF–HF statistics, an empirically validated assumption, see Figures 2a and 2b.

**Dual-path embeddings.** We split $X$ with an ideal band-pass filter, $L = \mathcal{F}_{\text{low}} X$, $H = \mathcal{F}_{\text{high}} X$, feed each band to the **same** encoder $\phi_\theta$ to obtain

$$\mathbf{z}_{content} = \phi_\theta(L), \quad \mathbf{z}_{noise} = \phi_\theta^{\text{HF}}(\mathsf{SRM}(H)),$$

and treat the softmaxed frames as empirical distributions $p(\mathbf{z}_{content})$, $p(\mathbf{z}_{noise})$.

**Alignment loss.** For label $y \in \{0 = \text{fake}, 1 = \text{real}\}$ we minimize

$$\mathcal{L}_{\text{align}} = y\, \mathrm{JS}\big[p(\mathbf{z}_{content}) \,\|\, p(\mathbf{z}_{noise})\big] + (1-y)\big(1 - \mathrm{JS}\big), \tag{1}$$

*pulling* LF and HF embeddings together for real speech and *pushing* them apart for fakes.

**Error bound.** Pinsker's inequality turns the Jensen–Shannon divergence $\mathcal{D}_{\text{JS}} = \mathrm{JS}[p(\mathbf{z}_{content}) \,\|\, p(\mathbf{z}_{noise})]$ into a Bayes error bound $P_e \leq \frac{1}{2}\sqrt{2\,\mathcal{D}_{\text{JS}}}$. Thus equation 1 **shrinks** $P_e$ for genuine pairs ($\mathcal{D}_{\text{JS}} \downarrow$) and **widens** the margin for fakes ($\mathcal{D}_{\text{JS}} \uparrow$), with the HF encoder acting as a targeted regularizer on the generator's weak band.

**The HF "hole"** is not merely an energy dip, it *breaks* the natural LF $\leftrightarrow$ HF dependency of real speech. SONAR restores this dependency for genuine audio and accentuates its absence for forgeries, turning a fundamental spectral-bias flaw into a reliable discriminative signal.

## 4 METHODOLOGY

**From Mathematical Motivation to Real-Life Application.** Our statistical analysis (Fig. 2) confirmed that fakes break the natural LF–HF co-modulation of real speech, supporting the factorization assumption $p_{\text{fake}}(L, H) \approx p(L)q(H)$. The mathematical motivation in Sec. 3 showed that aligning LF and HF embeddings via the JS loss directly tightens the Bayes error bound. SONAR operationalizes this by: (i) splitting each input $X$ into content ($L$) and noise ($H$) with an SRM-constrained filter bank, (ii) encoding both with twin XLSR paths to obtain $\mathbf{z}_c, \mathbf{z}_n$, and (iii) regulating their JS divergence through equation 1. For reals the loss minimizes divergence, restoring LF–HF dependency, for fakes it maximizes it, amplifying artifacts and widening the decision margin predicted by theory. We implement this dual-path head on AASIST Tak et al. (2022b), and in SONAR-Finetune attach it to XLSR-Mamba Xiao & Das (2024), training only the frequency-enhancing head for efficiency.

### 4.1 SONAR FEATURE EXTRACTION

The SONAR architecture consists of:

- **Content Feature Extractor (CFE):** A Wav2Vec2.0 XLSR Encoder.
- **Noise Feature Extractor (NFE):** A module based on constrained SRM filters followed by a Wav2Vec2.0 XLSR Encoder.

- **Fusion via Cross-Attention:** This merges the two feature streams into a unified representation.

Figure 1 illustrates the overall system.

### 4.1.1 CONTENT FEATURE EXTRACTION (CFE)

Given an input signal $\mathbf{x} \in \mathbb{R}^T$, we extract content features as:

$$\mathbf{z}_{\text{content}} = \text{CFE}(\mathbf{x}) \in \mathbb{R}^{F \times D}, \tag{2}$$

with $F$ time steps and feature dimension $D$.

### 4.1.2 NOISE FEATURE EXTRACTION (NFE)

The Noise Feature Extractor (NFE) builds on a Rich Feature Extraction (**RFE**) module (Figure 3). This module leverages constrained SRM filters to emphasize high-frequency components, which are then processed by the Wav2Vec2.0 XLSR Encoder Babu et al. (2021). The XLSR encoder weights were *not shared* across branches, allowing the model to learn disentangled representations of noise and content. A discussion of the resulting computational cost is provided in Appendix A.

**Constrained SRM Filters:** We initialize $M$ (hyperparameter for number of filters) learnable filters, each of length 5, with two key constraints:

$$w_i[m] = -1, \quad \text{(central coefficient)} \tag{3}$$

$$\sum_k w_i[k] = 0, \quad \text{(zero-sum constraint)} \tag{4}$$

where $w_i[m]$ is the $i$-th filter at index $m$ and we initialize the weights from $N(0, I)$. To ensure these are *hard constraints*, after every optimizer step we project the filters back to the constraint set: each filter is divided by the negative of its center coefficient (fixing the middle entry to $-1$), and then its mean is subtracted to enforce strict zero-sum. This guarantees that the constraints hold exactly throughout training without requiring reparameterization or relaxation.

These enforced constraints force each filter to act as a high-pass operator, suppressing low-frequency (content) information and highlighting high-frequency noise, consistent with prior work Bayar & Stamm (2016); Zhu et al. (2024); Han et al. (2021). Given an input signal $\mathbf{x}$, convolving with these filters yields:

$$\mathbf{F}_{\text{noise}} = \text{Conv1D}_{\text{SRM}}(\mathbf{x}), \tag{5}$$

where $\mathbf{F}_{\text{noise}} \in \mathbb{R}^{M \times T}$ contains the intermediate noise feature maps.

Extracting these high frequency features enables the network to detect subtle discrepancies in fake audio.

### 4.1.3 FUSION AND CLASSIFICATION

The content and noise embeddings, $\mathbf{z}_{\text{content}}$ and $\mathbf{z}_{\text{noise}}$, are fused using a cross-attention mechanism. We conducted all experiments with an embedding dimension of 1024 and 8 attention heads.

$$\mathbf{e}_{\text{out}} = \text{CA}(\mathbf{z}_{\text{content}}, \mathbf{z}_{\text{noise}}) \in \mathbb{R}^{F \times D}. \tag{6}$$

The fused representation is then fed to the AASIST classifier Jung et al. (2022):

$$\hat{y} = \text{AASIST}(\mathbf{e}_{\text{out}}) \in \mathbb{R}^2, \tag{7}$$

where $\hat{y}$ is the score vector for the real/fake decision.

The architecture of XLSR and AASIST models are detailed in the appendix.

## 4.2 Training Objective

### 4.2.1 JS-Based Loss for Real vs. Spoof embedded frequency distributions

Let $\mathbf{Z}_{\text{content}} \in \mathbb{R}^{F \times D}$ and $\mathbf{Z}_{\text{noise}} \in \mathbb{R}^{F \times D}$ denote the content and noise embeddings extracted from the dual path feature extractor for an audio example $\mathbf{x}$. Our goal is to increases the probability distance between the frequency embeddings of fake data while simultaneously reducing the distance among those of real data.

To achieve this, we employ the Jensen–Shannon (JS) divergence Fuglede & Topsøe (2004) as a metric for comparing distributions. First, we apply a frame-wise softmax to each embedding, converting them into probability distributions. Then, we compute a single JS divergence score, $\text{JS}\big(\mathbf{z}_{\text{content}}, \mathbf{z}_{\text{noise}}\big)$, using $\log_2$ as the logarithmic base. This ensures that the divergence is normalized to the range $[0, 1]$, facilitating stable comparison across samples.

**Frame-wise JS Divergence:** At each frame $i$, we can treat $\mathbf{z}_{\text{content}}[i]$ and $\mathbf{z}_{\text{noise}}[i]$ as two embeddings in $\mathbb{R}^D$. By applying $\mathrm{softmax}$ to each and get two discrete distributions $\mathbf{p}_{\text{content}}[i]$ and $\mathbf{p}_{\text{noise}}[i]$, where the final score for audio $\mathbf{x}$ with embeddings $\mathbf{Z}_{\text{content}}, \mathbf{Z}_{\text{noise}}$ will be:

$$\text{JS}\big(\mathbf{Z}_{\text{content}}, \mathbf{Z}_{\text{noise}}\big) \;=\; \frac{1}{F} \sum_{i=1}^{F} \text{JS}\big(\mathbf{p}_{\text{content}}[i], \; \mathbf{p}_{\text{noise}}[i]\big)$$

We label the example with $y = 1$ if it is *real*, and $y = 0$ if it is *spoof*.

**JS-Based Loss:**

With these settings, we define the custom loss function for each sample $(x, y)$:

Where $\mathbf{Z}_{\text{content}}, \mathbf{Z}_{\text{noise}}$ are the embeddings from our **SONAR** feature extraction.

$$L_{\text{JS}}(x, y) = y \cdot \text{JS}(\mathbf{z}_c, \mathbf{z}_n) + (1 - y) \cdot \big(1 - \text{JS}(\mathbf{z}_c, \mathbf{z}_n)\big) \tag{8}$$

We combine the above $L_{\text{JS}}$ (align loss as in equation 1) with a weighted cross-entropy (WCE) loss, which handles the real/fake classification in a more conventional way and accounts for class imbalance:

$$\mathcal{L}(x, y) \;=\; \text{WCE}\big(\hat{y}, \, y\big) \;+\; \lambda_{JS} \cdot L_{\text{JS}}\big(\mathbf{z}_{\text{content}}, \mathbf{z}_{\text{noise}}\big), \tag{9}$$

where $\hat{y} \in [0, 1]$ is the classifier's predicted probability of being real. The scalar $\lambda_{JS}$ balances how strongly the network must enforce the JS-based criterion. After ablation study, we chose to be $\lambda_{JS}{=}1$.

## 4.3 SONAR-Lite Setup

We propose SONAR-Lite to evaluate the intrinsic discriminative power of our frequency-guided backbone. Replacing AASIST with a lightweight two-layer MLP isolates the contribution of the dual-path features, which alone suffice to robustly separate real and spoofed inputs. Despite its simplicity, SONAR-Lite still attains state-of-the-art performance. Let $\mathbf{z}_{\text{content}}, \mathbf{z}_{\text{noise}} \in \mathbb{R}^{B \times T \times D}$ denote the dual-path embeddings, aggregated by mean pooling over time:

$$\tilde{\mathbf{z}}_{\text{content}} = \frac{1}{T} \sum_{t=1}^{T} \mathbf{z}_{\text{content},t}, \quad \tilde{\mathbf{z}}_{\text{noise}} = \frac{1}{T} \sum_{t=1}^{T} \mathbf{z}_{\text{noise},t},$$

and concatenated into a single vector $\mathbf{x} = [\tilde{\mathbf{z}}_{\text{content}}, \tilde{\mathbf{z}}_{\text{noise}}]$. This vector is then fed into a lightweight two-layer fully connected classifier.

This simplified design demonstrates that the dual path features alone provide robust discriminative power, as validated by our experimental results (see Table 1).

## 4.4 SONAR-Finetune

To address the computational overhead of training dual XLSR encoders from scratch, we explored a finetuning strategy leveraging the pre-trained XLSR-Mamba pipeline Xiao & Das (2024). Under the

| Model | LA↓ | DF↓ | ITW↓ |
|---|---|---|---|
| WavLM-Large+MFA Guo et al. (2024) | 5.08 | 2.56 | – |
| XLSR+AASIST Tak et al. (2022b) | 1.90 | 3.69 | 10.46 |
| XLSR+MoE Wang et al. (2024) | – | – | 9.17 |
| XLSR+Conformer Rosello et al. (2023) | 0.97 | 2.58 | 8.42 |
| XLSR+Conformer+TCM Truong et al. (2024) | 1.18 | 2.25 | 7.79 |
| XLSR-SLS Zhang et al. (2024) | 2.87 | 1.92 | 7.46 |
| XLSR-Mamba Xiao & Das (2024) | **0.93** | 1.88 | 6.71 |
| **SONAR**-Lite (M=30, $\lambda_{JS} = 1$) | 1.78 (2.03) | 2.11 (2.5) | 6.98 (7.2) |
| **SONAR**-Full (M=30, $\lambda_{JS} = 1$, enhancing Tak et al. (2022b)) | 1.55 (1.68) | 1.57 (1.95) | 6.00 (6.8) |
| **SONAR**-Finetune (M=30, $\lambda_{JS} = 1$) on Xiao & Das (2024) | 1.20 (1.30) | **1.45** (1.62) | **5.43** (5.8) |

Table 1: EER (%) on ASVspoof 2021 LA, DF, and In The Wild datasets. Bold entries are best per column. Results from prior work are single-run values, SONAR variants report best (mean) over 3 runs. Statistical significance: SONAR-Full vs. AASIST on ITW ($t = 19.4$, $p = 0.0026$) and SONAR-Finetune vs. XLSR-Mamba on ITW ($t = 4.73$, $p = 0.0419$) both confirm robust improvements ($p < 0.05$).

assumption that the original XLSR model, inherently favors low frequency (content) information, we designate it as the content path. To complement this, We then insert our NFE module into the pipeline to extract complementary high frequency cues. These content and noise embeddings are fused via the cross-attention and passed into the mostly-frozen Mamba classifier, with only the noise XLSR, fusion module, and the final two Mamba layers updated during training. The training process started from the best reported checkpoint of the paper and was stopped once the alignment loss (1) plateaued, which in our experiments consistently occurred within 6 epochs. This efficient setup allows us to retain most of the pre-trained model's capacity while enriching it with high frequency information with low effort. Remarkably, this finetuned configuration achieves state-of-the-art performance, as detailed in Table 1.

## 5 EXPERIMENTS & RESULTS

### 5.1 DATASETS & TRAINING CONFIGURATION

For our experiments, in the same manner that all audio DF detection models are trained, we utilized the ASVspoof 2019 Logical Access (LA) training set Yamagishi et al. (2019) for model training and the corresponding LA validation set for tuning. The model wasn't expose to any other data set in the process. For evaluation, we used the ASVspoof 2021 competition datasets Delgado et al. (2021) that were designed only for testing. They are covering both Logical Access (clean TTS and VC) and Deep Fake scenarios. Since the ASVspoof 2019 LA training set is highly imbalanced (approximately 1:9 real to fake ratio), we employed a *weighted cross-entropy* (WCE) loss with class priors estimated from the training split. This ensured that the model avoided bias toward the majority class during optimization. To further assess generalization, we also evaluated on the In The Wild dataset introduced by Müller et al. (2022), which contains diverse, real world audio samples. This comprehensive protocol ensures robustness in both controlled and uncontrolled environments. Audio data were segmented into approximately 4-second clips (64,600 samples).As in prior work Tak et al. (2022b); Xiao & Das (2024); Rosello et al. (2023), we applied RawBoost Tak et al. (2022a). We tested several setups with consistent trends and adopted the standard config for fairness. SONAR-Full and SONAR-Lite training we optimized our model using the AdamW optimizer with an learning rate of $10^{-5}$, decaying to $10^{-8}$ via cosine annealing. For the finetune version we did the same but only optimized the NFE, fusion layer, and the last two layers of the mamba classifier. For speed purposes, training was performed on 4 NVIDIA L40 GPUs with an effective batch size of $28 \times 4$. The proposed model can fit inside a single L-40. Each experiment was run three times with different random seeds, employing early stopping (patience of three epochs) and selecting the model with the lowest validation EER. All results are reproducible using our open source code that will be released upon acceptance. Audio deepfake detection is evaluated using the **Equal Error Rate (EER)**, which is the point where the false acceptance rate equals the false rejection rate. Despite

| Method / Augmentation | DF ↓ | LA ↓ | ITW ↓ |
|---|---|---|---|
| *Architectural Ablations* | | | |
| SONAR-Full w/o RFE, JS | 2.54 | 2.93 | 8.91 |
| SONAR-Full w/o RFE | 2.40 | 2.48 | 8.44 |
| SONAR-Full w/o JS | 2.65 | 2.90 | 8.50 |
| SONAR-Full w/ non-learnable RFE (M=30) | 2.30 | 2.36 | 8.20 |
| SONAR-Full w/ $M=1$ SRM | 2.83 | 2.91 | 8.00 |
| SONAR-Full w/ $M=10$ SRM | 2.43 | 2.51 | 7.40 |
| SONAR-Full w/ $\lambda_{JS}=0.5$ | 2.42 | 2.61 | 7.91 |
| SONAR-Full w/ $\lambda_{JS}=0.8$ | 1.90 | 1.78 | 7.02 |
| SONAR-Full w/ SRM and $\lambda_{JS}=1$ (best configuration) | **1.57** | **1.55** | **6.00** |
| *Robustness to Sampling Rates / Codecs (ITW only)* | | | |
| Resample → 44.1 kHz | | | $\approx 0$ |
| Resample → 48 kHz | | | $\approx 0$ |
| MP3 (64 kbps) | | | medium jitter ($|\Delta p| \leq 0.1$) |
| Opus (32 kbps) | | | medium jitter ($|\Delta p| \leq 0.1$) |
| Vorbis (q3) | | | small jitter ($|\Delta p| \leq 0.05$) |

Table 2: **Ablation study (SONAR-Full).** Top: pooled EER (%) on DF, LA, and ITW sets under different architectural ablations. Bottom: robustness analysis of SONAR-Full (trained on the standard ASVspoof2019 dataset at 16 kHz) evaluated on the ITW test set becuase its difficulty under different resampling and codec augmentations. Results are reported as probability shifts in the softmax outputs.

having nearly double the number of parameters, SONAR-Full introduces only a marginal increase in inference time compared to latest SOTA XLSR-Mamba Xiao & Das (2024), as shown in Fig. 6.

## 5.2 RESULTS

**SONAR** achieves new state-of-the-art performance across DF, LA, and IN THE WILD (Table 1):

- **DF:** *SONAR-Full* and *SONAR-Finetune* reach **1.57%** and **1.45%**, surpassing all prior methods.

- **LA:** Both achieve competitive performance (**1.20%–1.55%**) and are the strongest models trained with a single run.

- **In The Wild:** *SONAR-Full* and *SONAR-Finetune* set a new benchmark at **6.00%** and **5.43%**.

The LA decrease against models like XLSR-Mamba Xiao & Das (2024) and XLSR-Conformer Truong et al. (2024),Rosello et al. (2023) is explained by a key evaluation difference: their reported results rely on *checkpoint averaging* or run-smoothing, while SONAR is evaluated strictly under **single training runs**. Our reported values are means over three independent seeds, ensuring that improvements reflect genuine convergence rather than post-hoc stabilization. This distinction naturally accounts for the apparent advantage of XLSR-Mamba on LA (0.93% vs. 1.55% for SONAR-Full). Under a fair single-run protocol, SONAR achieves SOTA performance on LA, DF and ITW. Thus, the small LA difference is not a weakness but an expected outcome of different evaluation protocols, and SONAR's frequency-guided alignment offers stronger and more robust generalization in out-of-distribution settings.

**Convergence speed.** SONAR also converges rapidly: while Tak et al. Tak et al. (2022b) trained for 100 epochs, SONAR-Full stabilizes in 12, and SONAR-Finetune in only 4–6. Despite the added branch, SONAR attains higher accuracy with nearly an order-of-magnitude faster training, thanks to the alignment loss (Eq. 8) which tightens LF–HF coupling and accelerates separation of real and fake embeddings.

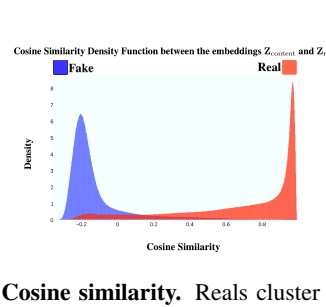

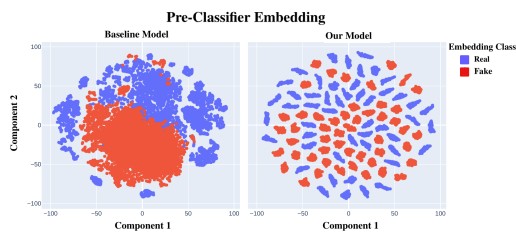

(a) **Cosine similarity.** Reals cluster near 1 (LF–HF aligned), while fakes center around $-0.2$, indicating dissimilarity.

(b) **t-SNE embeddings.** SONAR-Full features yield clear separation between real and fake samples, unlike the single-XLSR baseline.

Figure 4: **Latent representation analysis of SONAR.** (a) t-SNE shows that SONAR's dual-path embeddings separate real and fake audio more distinctly than the baseline. (b) Cosine similarity histograms confirm that real speech preserves LF–HF coupling, while fakes exhibit disjoint embeddings.

### 5.3 ABLATION & ANALYSIS

Table 2 summarizes the ablation results for **SONAR-Full**. Both the RFE and JS alignment are essential: removing either leads to clear drops, with the absence of JS alone being particularly harmful on LA. Interestingly, removing both RFE and JS together performs slightly better than removing JS alone, since the simplified architecture avoids unstable, unconstrained noise embeddings. Varying $M$ and $\lambda_{JS}$ shows consistent robustness, with $\lambda_{JS}=1$ yielding the strongest overall results in the ablation. The codec and resampling tests further confirm that SONAR remains stable under common degradations what is making it practical usability.

**Embedding Analysis.** To assess latent separation, we evaluated SONAR-Full on the In-The-Wild set. Figure 4 shows that the alignment loss preserves strong LF–HF coupling in real speech (cosine similarity near 1), while fakes collapse toward negative values ($\approx -0.2$). t-SNE further reveals a much clearer real/fake split than the XLSR baseline, confirming that disentangling and aligning content and noise yields a more discriminative latent space.

## 6 CONCLUSION

**SONAR** reframes audio–deepfake detection as a *frequency-guided, contrastive* representation task. By splitting speech into complementary low- and high-frequency paths and regulating their latent divergence with a Jensen–Shannon loss, SONAR turns the generator's persistent "HF hole" into a decisive discriminative cue. Entirely data-driven, it requires no hand-crafted filters and offers a new lens for robust deepfake detection.

Empirically, SONAR-FULL and SONAR-FINETUNE achieve single-run **state-of-the-art** EERs with convergence up to **8× faster** than reported in strong baselines, while maintaining robustness to codecs and realistic bandwidth shifts. **SONAR** transforms spectral bias into a discriminative signal through a frequency-contrastive dual-path design, setting a new standard for audio deepfake detection. **Key Breakthroughs.**

- **SONAR: frequency-contrastive dual path.** The first audio deepfake detector to jointly model low-frequency content and high-frequency residuals, transforming spectral bias into a discriminative signal.

- **Learnable SRM & JS alignment.** A learnable SRM filter bank with a Jensen–Shannon loss explicitly aligns LF–HF embeddings for real audio while repelling fake pairs, producing disjoint latent manifolds.

- **State-of-the-art with fast convergence.** SONAR sets new SOTA with EERs of 1.57% (DF), 6.00% (ITW), and 1.55% (LA) in 12 epochs, while SONAR-Finetune improves further to 1.45% / 5.43% (DF/ITW) and 1.20% (LA) in 4 epochs. It is the best single-run system on LA and the overall SOTA on DF and In-the-Wild.

## REPRODUCIBILITY STATEMENT

We have taken several steps to ensure that the findings reported in this paper are reproducible. All datasets used are publicly available: ASVspoof 2019 and 2021 (Sec. 5.1, 4) and the In-the-Wild corpus Müller et al. (2022), 4. Our preprocessing and segmentation procedure (4-second clips, Default settings of RawBoost augmentation) is described in Sec. 5.1 and the source in 4. The full architecture of SONAR, including the constrained SRM filters, dual-path XLSR encoders, and cross-attention fusion, is detailed in Sec. 4, with additional implementation details provided in the Appendix (A). Training objectives, including the Jensen–Shannon alignment loss and weighted cross-entropy, are specified in Eq. 1–8. All experiments were run three times with different seeds, and statistical significance tests are reported in Table 1. An anonymous code repository containing source code training scripts accompanies this submission in the supplementary materials.

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

## A  APPENDIX

## XLSR ARCHITECTURE OVERVIEW

**XLSR** (Cross-Lingual Speech Representations) is a large-scale multilingual model based on the **Wav2Vec 2.0** architecture, trained on over 400,000 hours of speech across 128 languages. It is designed to learn universal speech representations that generalize well across languages and tasks. XLSR extends Wav2Vec 2.0 with larger model capacity and multilingual pretraining.

### CORE COMPONENTS

- **Feature Encoder.** The input waveform $x \in \mathbb{R}^T$ is first passed through a series of temporal convolutional layers that output a latent representation:

$$z = \text{ConvEncoder}(x) \in \mathbb{R}^{T' \times d}$$

  where $T' \ll T$ due to downsampling, and $d$ is the channel dimension.

- **Quantization Module (Pretraining Only).** A Gumbel-softmax quantizer maps $z$ to discrete latent codes $q(z)$ sampled from a learned codebook. This discrete representation is used as a target in contrastive learning. The quantization module is discarded after pretraining.

- **Transformer Encoder.** The encoded sequence $z$ is fed into a multi-layer Transformer:

$$c = \text{Transformer}(z)$$

  For XLSR 300M, the Transformer consists of 24 layers, each with hidden size 1024, 16 self-attention heads, and feed-forward networks of dimension 4096.

- **Contrastive Objective (Pretraining).** The model is trained to distinguish the true quantized target $q(z)$ from a set of distractors using a contrastive loss, encouraging the model to learn meaningful representations without labels.

### DOWNSTREAM USAGE

After pretraining, the quantizer and contrastive heads are removed. The contextualized features $c$ are used as inputs to downstream tasks such as speech recognition, speaker verification, or deepfake detection. In our work, we extract $c$ either in frozen mode or via finetuning, and feed it into a task-specific classifier.

## AASIST ARCHITECTURE OVERVIEW

**AASIST** (Audio Anti-Spoofing using Integrated Spectra-Temporal Modeling) is a deep learning model designed for detecting spoofed audio in speaker verification systems. It combines spectra-temporal modeling with attention-based mechanisms to robustly capture discriminative features between genuine and fake audio, particularly under real world conditions.

### CORE COMPONENTS

- **Learnable Frontend:** The raw waveform $x \in \mathbb{R}^T$ is first passed through a 1D convolutional frontend that acts as a learnable filterbank:

$$x_{\text{spec}} = \text{Conv1D}(x)$$

  This mimics handcrafted feature extraction (e.g., STFT or filterbanks) in a data-driven way and outputs time-frequency like representations.

- **Graph Attention Layer (GAT):** The core innovation of AASIST is to treat the spectro-temporal representation as a graph where each node corresponds to a time-frequency patch. A Graph Attention Network (GAT) models the structured relationships between these patches:

$$h'_i = \sum_{j \in \mathcal{N}(i)} \alpha_{ij} \mathbf{W} h_j$$

where $\alpha_{ij}$ are attention weights learned over neighbors $\mathcal{N}(i)$, and $\mathbf{W}$ is a shared linear transform.

- **Spectro-Temporal Blocks:** A series of convolutional blocks capture local patterns in both time and frequency domains. These are alternated with GAT layers to jointly model local and global context.

- **Global Aggregation and Classification:** After the GAT and convolutional layers, the model aggregates features via global average pooling and passes them through fully connected layers for binary classification:

$$\hat{y} = \sigma(\text{MLP}(\text{GAP}(H)))$$

### ADVANTAGES

- **Spectro-Temporal Awareness:** By combining CNNs and GATs, AASIST captures both fine-grained local patterns and long-range spectral dependencies.

- **Fully Learnable Pipeline:** From waveform to classification, the architecture is end-to-end trainable without handcrafted features.

- **Strong Benchmarks:** AASIST achieves state-of-the-art performance on ASVspoof 2019 and 2021 logical access (LA) and deepfake (DF) subsets, especially under noisy and real world conditions.

### USAGE IN OUR WORK

We adopt AASIST as a strong baseline in our experiments on **SONAR-Full** model.Its ability to detect both TTS and VC-based attacks makes it a competitive model for evaluating deepfake detection methods.

### LIMITATIONS

**Sensitivity to resampling and reliance on high-frequency cues.** Figure 5 reveals that the Equal-Error Rate increases monotonically when the input audio is down-sampled for example, from 16 kHz to 4 kHz, thereby stripping energy above the new Nyquist frequency. The degradation from a state-of-the-art 6demonstrates that SONAR exploits high-frequency noise artifacts introduced during deep-fake synthesis. While this helps on clean, full-bandwidth recordings, it also exposes a limitation: the detector becomes less robust when real-world pipelines or codecs apply aggressive low-pass filtering or resampling. However, we note that most audio found *in the wild* is sampled at 16,kHz or higher, meaning this sensitivity is less likely to affect practical deployments. Moreover, our model maintained strong performance across a range of common codecs, including high-quality MP3 compression, indicating robustness to realistic encoding conditions. Practitioners should therefore (i) preserve as high a capture sample-rate as feasible, or (ii) retrain / fine-tune the model on data that reflect the target bandwidth and compression conditions. **Model size and compute.** Although the dual-path design roughly doubles the parameter count to 650 M (with XLSR large), it remains feasible to train for 12 epochs on a single L40 GPU standard for real world remote server deployments. We leave further optimization via parameter sharing and pruning for future work.

**Modality scope.** Experiments are confined to audio. While the frequency-guided principle is generic, porting SONAR to images or video will require modality-specific high-pass filtering and fusion schemes, which we have not yet explored.

**Dataset coverage.** Evaluation spans ASVspoof 2021 (LA/DF) and the Müller *in-the-wild* corpus, although these are the academic benchmarks for spoofing detection, unseen spoof mechanisms or languages may still degrade performance.

**False positives/negatives.** Like any detector, SONAR can misclassify highly compressed real speech or exceptionally well-crafted fakes, which could erode user trust, threshold calibration for different deployment domains remains an open question.

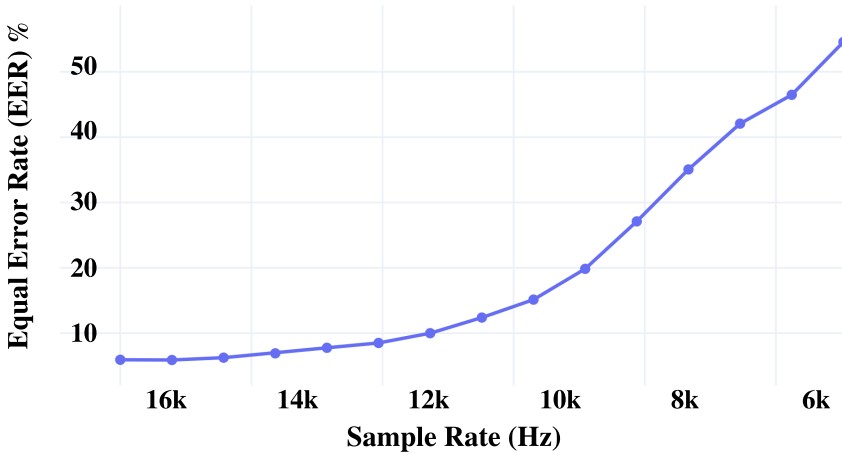

**Model EER (%) On In-The-wild Test Set With Different Sample Rates**

Figure 5: Impact of resampling on detection accuracy. Equal-Error Rate (EER) rises as the sampling rate (SR) of the test set is lowered, confirming that the model relies on high-frequency artifacts introduced during deep-fake synthesis.

| Model | License | URL |
|---|---|---|
| XLSR (fairseq) | MIT | `https://github.com/facebookresearch/fairseq` |
| XLSR-Mamba | MIT | `https://github.com/swagshaw/XLSR-Mamba` |
| AASIST | MIT | `https://github.com/clovaai/aasist` |

Table 3: Licenses for pretrained models.

## LICENSING OF THIRD-PARTY ASSETS

All third-party assets used in this work, including models and datasets, are listed in Table 3, along with their license terms and usage conditions. We ensure that all included components comply with their respective open source or research use licenses.

All third-party assets used in this work are listed below, including pretrained models and datasets, along with their license terms and URLs.

## BROADER IMPACT

The rapid commoditisation of neural voice cloning poses concrete risks in day-to-day life from account-takeover attempts at banks and call-centre fraud, to automated disinformation in political campaigns and social media. SONAR contributes a stronger line of defence: it is a *detection-only* model that neither synthesises nor enhances fake audio. Wider deployment could therefore help journalists, financial institutions and platform moderators to flag spoofed content early, limiting downstream harm. On the negative side, ever-stronger detectors may escalate an adversarial arms-race, encouraging attackers to craft subtler manipulations. We mitigate this by (i) releasing code and eval-

| Dataset | License | URL / Terms |
|---|---|---|
| In The Wild | Apache 2.0 | `https:// deepfake-total. com/in_the_ wild` |
| ASVspoof (LA/DF) | 2019 ODC-By v1.0 | `https:// datashare.ed. ac.uk/handle/ 10283/3336` |
| ASVspoof (LA/DF) | 2021 ODC-By v1.0 | `https: //doi.org/10. 5281/zenodo. 4837263` |

Table 4: Licenses for datasets.

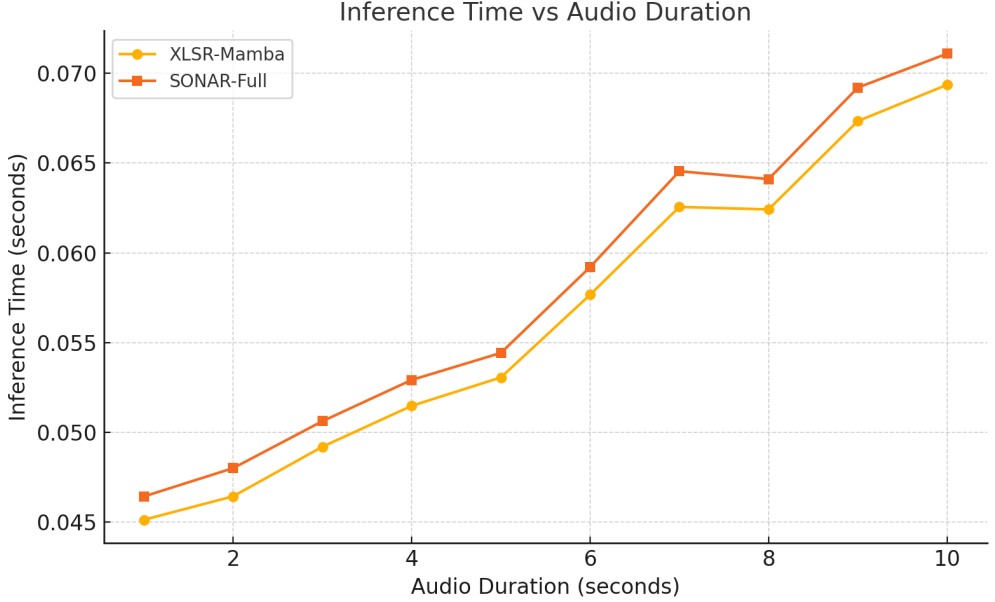

Figure 6: **Inference latency scales linearly with audio length.** We compare inference times (in seconds) for the XLSR-Mamba and SONAR-Full models across increasing audio durations from 1 to 10 seconds. SONAR introduces only a minimal overhead relative to XLSR-Mamba, while delivering improved detection performance (cf. Table 1).

uation scripts to foster transparent benchmarks, and (ii) encouraging periodic re-training on newly emerging spoof methods. The model uses only publicly available speech data collected with consent, and stores no personal attributes beyond the embeddings required for classification.

We build directly on the publicly released AASIST and XLS-R reference implementations, adopting the CUDA-optimised training framework of Tak et al. All experiments were run end-to-end on a single NVIDIA L40 (48 GB) GPU under PyTorch 2.2 with CUDA 12.2. The complete source code, Hydra configs, pretrained checkpoints, and the shell scripts used to reproduce every table and figure accompany this paper in the supplementary package.

COMPUTATIONAL COST.

We analyzed the additional cost of SONAR relative to a single-stream XLSR baseline. The extra components are: (i) the Rich Feature Extractor (RFE), (ii) a second encoder branch, and (iii) a cross-attention fusion.

**RFE and fusion are negligible.** For a $4\,$s clip at $16\,$kHz with $M{=}10$ filters, the RFE adds only $\sim$8M FLOPs ($< 0.01$G), and the cross-attention adds $\sim$0.16G FLOPs. Both are $< 1\%$ of a single XLSR pass.

**Encoders dominate.** SONAR-Full essentially doubles the encoder cost, giving $\approx 2\times$ the parameters and FLOPs of XLSR. However, since the two streams run concurrently on GPU, the measured wall-clock latency increases by only $15$–$25\%$ (Fig. 6), not $100\%$.

