# OpenReview forum: "SONAR: Spectral‑Contrastive Audio Residuals for Robust Deepfake Detection"
_ICLR.cc/2026/Conference — ICLR 2026 Conference Withdrawn Submission_

### Official Review · Reviewer_FEJa · 2025-10-15

**Soundness:** 2
**Presentation:** 2
**Contribution:** 1
**Rating:** 2
**Confidence:** 5

**Summary:**

This paper proposes a simple method that performs frequency-differentiated learning using a dual-branch architecture.

**Strengths:**

The motivation of this paper is good, and the proposed method has achieved certain results.

**Weaknesses:**

Although the approach achieves some results, I have several concerns:

1. The proposed method lacks originality — it mainly stacks existing techniques without introducing any novel design.
2. From the performance perspective, the difference between *XLSR-Mamba* and the proposed method in Table 1 is not significant.
3. More importantly, in current speech deepfake detection models, the majority of parameters are concentrated in the front-end feature extractor. Since this work employs two such extractors, the training convergence naturally becomes faster. Therefore, the paper should thoroughly discuss the parameter count and computational cost.
4. The paper claims to emphasize *generalization ability*—that is, the capacity to handle out-of-domain data—but the title focuses on *robustness in speech deepfake detection*, which is confusing and inconsistent.
5. The phenomenon of spectral bias arises because high-frequency regions tend to reveal *shallow* forgery artifacts (e.g., high-frequency distortion or spectral loss), yet these cues have poor generalization. In contrast, low-frequency regions often capture *deeper* forgery cues that generalize better. Thus, explicitly guiding the model toward low-frequency learning can also be effective, and many related approaches already exist.

**Questions:**

Refer to the weaknesses listed above.

---

### Official Review · Reviewer_atH9 · 2025-10-29

**Soundness:** 3
**Presentation:** 2
**Contribution:** 3
**Rating:** 6
**Confidence:** 4

**Summary:**

This work introduces SONAR (Spectral-cONtrastive Audio Residuals), a deepfake detector that operates on the principle that the statistical coupling between low-frequency (LF) content and high-frequency (HF) residuals is a key differentiator between real and fake audio. The method integrates two core components: 1) a Dual-Path Architecture is employed to explicitly disentangle an audio signal into separate LF content and HF residual representations; a key element here is a learnable SRM filter bank that isolates the HF components; 2) a Jensen-Shannon (JS) divergence-based loss contrastively regulates the relationship between the two representations, forcing the LF-HF embeddings of real audio to be statistically similar while forcing those of fakes to be dissimilar. The empirical results show the effectiveness of the proposed approaches.

**Strengths:**

1. The work's primary contribution is a novel detection paradigm. It is the first to jointly model the statistical relationship between low-frequency (LF) content and high-frequency (HF) residuals.
2. The framework employs learnable SRM filters to isolate HF residuals, then uses a JS divergence-based loss to contrastively align LF-HF embeddings for real audio while repelling them for fakes—a statistical consistency learning mechanism unseen in prior work.

**Weaknesses:**

Despite mitigated inference latency, the model's doubled size and high computational cost present a barrier to practical training and deployment.

**Questions:**

There is a more recent ASVspoof5 (2024) competition, how does the model compare with models in this new competition?

---

### Official Review · Reviewer_pNNK · 2025-10-31

**Soundness:** 3
**Presentation:** 2
**Contribution:** 3
**Rating:** 4
**Confidence:** 2

**Summary:**

This paper introduces a dual-path framework equipped with a novel Jensen–Shannon contrastive loss that boosts the OOD generalization of deepfake audio detectors. SONAR explicitly disentangles and contrasts low-frequency and high-frequency features to counteract the spectral bias of neural networks.

**Strengths:**

Theoretical motivation: The design, rooted in the statistical coupling defects between LF and HF bands, offers deep insight.

**Weaknesses:**

1. In the SONAR-Full setup, two complete XLSR encoders must be trained simultaneously, which markedly raises training-time compute and memory demands. Although the paper proposes SONAR-Finetune which training only the frequency-augmentation heads to mitigate this, the efficiency of the full dual-encoder architecture remains a concern.
2. The study does not include a controlled ablation showing that the proposed learnable RFE outperforms a simple fixed high-pass filter. Consequently, the central innovation of learnability remains insufficiently demonstrated.
3. There is a lack of empirical analysis on how the Jensen–Shannon contrastive loss efficiently and robustly measures the joint distribution of LF/HF embeddings.

**Questions:**

1. The paper claims SOTA performance, but the parallel use of two full XLSR encoders introduces a massive increase in computational complexity and memory footprint during both training and inference. Please provide a detailed comparison of the Total FLOPs and Inference Latency of SONAR-Full against the single-stream XLSR baseline and the SOTA baselines. The current cost analysis (Table 4) is insufficient as it only shows the number of parameters.
2. The paper must include a direct ablation study comparing the performance of the learnable RFE against a simpler, fixed high-pass filter.
3. The core assumption is that all DF generators result in a similar statistical decoupling that the JS loss can detect. How does SONAR perform against novel or unseen generation methods that are designed to specifically minimize this LF/HF coupling defect?

---

### Official Review · Reviewer_rjAP · 2025-11-02

**Soundness:** 3
**Presentation:** 2
**Contribution:** 2
**Rating:** 6
**Confidence:** 3

**Summary:**

This paper proposes a method to improve the detection of modeling artifacts in the high-frequency regions of audio deepfake signals. The motivation is that most audio deepfake models tend to model low-frequency components first, leading to noticeable imperfections in the high-frequency range. To address this, the authors design a model with separate branches for low-, high-frequency components, and combine their outputs for the final prediction. In particular, the high-frequency branch employs a learnable high-pass filter to better capture fine-grained spectral artifacts. The performance was evaluated on two datasets which are ASVspoof 2021 and in-the-wild. And, the proposed method achieved results comparable to or better than state-of-the-art approaches. Overall, the paper provides a meaningful contribution to the exploration of methods that explicitly model high-frequency characteristics of deepfake audio. However, detailed analysis is somewhat missing.

**Strengths:**

The use of SRM filter and JS-based loss seems promising. The experimental setup and ablation study verified the usefulness of the proposed method.

**Weaknesses:**

Please note what SRM is from line 043, p.1.

In line 206, "for fakes it maximizes it".

Figure 4, the caption of (a) and (b) seems reverted.

**Questions:**

I'm also wondering if it's possible to show the probing result of each content and noise branches features. X-axis as a epoch, and the Y-axis as a accuracy of the probing result of these features might give further insights of the proposed method.

---

### Note · Authors · 2025-11-14

I have read and agree with the venue's withdrawal policy on behalf of myself and my co-authors.